# PAs Regulate Early Somatic Embryo Development by Changing the Gene Expression Level and the Hormonal Balance in *Dimocarpus longan* Lour.

**DOI:** 10.3390/genes13020317

**Published:** 2022-02-08

**Authors:** Chunwang Lai, Xiaojuan Zhou, Shuting Zhang, Xueying Zhang, Mengyu Liu, Chunyu Zhang, Xiaoqiong Xu, Xiaoping Xu, Xiaohui Chen, Yan Chen, Wenzhong Lin, Zhongxiong Lai, Yuling Lin

**Affiliations:** 1Institute of Horticultural Biotechnology, Fujian Agriculture and Forestry University, Fuzhou 350002, China; laichunwang@163.com (C.L.); shutingvictory@yeah.net (S.Z.); zxyyingg@163.com (X.Z.); lmy20210807@163.com (M.L.); zcynhba@163.com (C.Z.); xxq990921@163.com (X.X.); byxxp310107@163.com (X.X.); 18950589675@163.com (X.C.); spencer_cy@163.com (Y.C.); laizx01@163.com (Z.L.); 2Ganzhou Agricultural and Rural Bureau, Ganzhou 341000, China; zhouxiaojuandna@163.com; 3Quanzhou Agricultural Science Research Institute, Quanzhou 362212, China; lwzfjqz@163.com

**Keywords:** longan, polyamines, ethylene, genome-wide identification, plant endogenous hormones, embryogenic callus, globular embryo

## Abstract

Polyamines (PAs) play an important regulatory role in many basic cellular processes and physiological and biochemical processes. However, there are few studies on the identification of PA biosynthesis and metabolism family members and the role of PAs in the transition of plant embryogenic calli (EC) into globular embryos (GE), especially in perennial woody plants. We identified 20 genes involved in PA biosynthesis and metabolism from the third-generation genome of longan (*Dimocarpus longan* Lour.). There were no significant differences between longan and other species regarding the number of members, and they had high similarity with *Citrus sinensis*. Light, plant hormones and a variety of stress *cis*-acting elements were found in these family members. The biosynthesis and metabolism of PAs in longan were mainly completed by *DlADC2*, *DlSAMDC2*, *DlSAMDC3*, *DlSPDS1A*, *DlSPMS*, *DlCuAOB*, *DlCuAO3A*, *DlPAO2* and *DlPAO4B*. In addition, 0.01 mmol∙L^−1^ 1-aminocyclopropane-1-carboxylic acid (ACC), putrescine (Put) and spermine (Spm), could promote the transformation of EC into GE, and Spm treatment had the best effect, while 0.01 mmol∙L^−1^ D-arginine (D-arg) treatment inhibited the process. The period between the 9th and 11th days was key for the transformation of EC into GE in longan. There were higher levels of gibberellin (GA), salicylic acid (SA) and abscisic acid (ABA) and lower levels of indole-3-acetic acid (IAA), ethylene and hydrogen peroxide (H_2_O_2_) in this key period. The expression levels in this period of *DlADC2*, *DlODC*, *DlSPDS1A*, *DlCuAOB* and *DlPAO4B* were upregulated, while those of *DlSAMDC2* and *DlSPMS* were downregulated. These results showed that the exogenous ACC, D-arg and PAs could regulate the transformation of EC into GE in longan by changing the content of endogenous hormones and the expression levels of PA biosynthesis and metabolism genes. This study provided a foundation for further determining the physicochemical properties and molecular evolution characteristics of the PA biosynthesis and metabolism gene families, and explored the mechanism of PAs and ethylene for regulating the transformation of plant EC into GE.

## 1. Introduction

Polyamines (PAs) are low molecular weight aliphatic nitrogenous bases with biological activity, widely existing in plants, animals and microorganisms [1]. The main PAs with important physiological functions in plants are putrescine (Put), spermidine (Spd) and spermine (Spm), which are found in cells as free amines or as amide conjugates [2]. In normal cells, PAs are protonated and can be bound to anions, such as deoxyribonucleic acid (DNA), ribonucleic acid (RNA), chromatin and protein, to stabilize the functional structure of macromolecules [3]. In addition, PAs act as regulators in many basic cellular processes, such as cell division and differentiation, cell death, synthesis and the functional expression of nucleic acids and proteins [4,5,6]. In plants, PAs are involved in many physiological and biochemical processes, such as xylem formation, organ development and senescence, seed formation, fruit ripening, etc. [7]. PAs are the important stress response factors, and are widely involved in a variety of biotic and abiotic stress responses in plants [8]. Their signal pathways are directly or indirectly related to many metabolic pathways in plants. Meanwhile, as signal molecules, PAs interact with complex hormone signal networks in plants, and participate in the response of plants to stress [9]. The biosynthesis and metabolism pathways of PAs have been identified in some plants, such as *Arabidopsis thaliana* (*Arabidopsis*) [10], apples [11] and peaches [12]. Put is the intermediate of PA synthesis. In most plants, Put biosynthesis involves two pathways. One is the arginine decarboxylase (ADC)-catalyzed reaction; the second is the ornithine decarboxylase (ODC)-catalyzed reaction. Another important substrate for PA synthesis is S-adenosylmethionine (SAM). It decarboxylates to produce decarboxylated S-adenosylmethionine (dcSAM) under the catalysis of S-adenosylmethionine decarboxylase (SAMDC). Under the catalysis of spermidine synthase (SPDS), Put and dcSAM produce Spd, while Spd and dcSAM produce Spm under the catalysis of Spm synthase (SPMS) [2,13,14]. The oxidation of PAs is mainly carried out by copper ammonia oxidase (CuAO) and PA oxidase (PAO), in which the oxidation of Put is carried out by CuAO, and the oxidation of Spd and Spm are carried out by PAO [12,15,16]. SAM is also an important substrate for ethylene synthesis. It is carboxylated to 1-aminocyclopropane-1-carboxylic acid (ACC) by 1-aminocyclopropane-1-carboxylic acid synthase (ACS) and, finally, ACC is oxidized to ethylene by 1-aminocyclopropane-1-carboxylic acid oxidase (ACO) [17]. The ability of plants to resist external stress can be achieved by reasonably increasing the content of PAs. Bound Put accumulation maintains the capacity of antioxidant enzymes and protects against Al-induced oxidative damage in wheat [8]. In cucumber plants, Spd could improve photosynthetic capacity and tolerance to salinity by regulating the gene expression and activity of key enzymes for CO_2_ fixation [18]. Accumulations of Put confer increased drought tolerance in the transgenic plants of tobacco (*Nicotiana tabacum*) and lemon (*Citrus lemon*) [19]. Additionally, when the synthesis of Put was inhibited, the drought resistance of plants decreased [20]. The results of Gong et al. show that Spd and Spm are more essential than Put for apples, especially when responding to alkaline or salt stress [11]. At present, there are many studies on the interaction between PAs and plant endogenous hormones. The regulation of PAs in strawberry fruit ripening is closely related to the levels of ABA, ethylene and IAA [7]. The external application of Spm inhibited ABA synthesis and improved plant reproductive health in osmotic-stressed soybean pods [21].

The somatic embryogenesis (SE) of plants needs to maintain a good balance of PA substances, and the increase in PA content is considered to be the premise of SE. The effect of plant hormones on SE are actually mediated by PAs, in which PAs play the role of “second messenger” [22]. Maintaining a high proportion of Put / (Spd + Spm) is conducive to the production of embryogenic calli (EC) [23]. It has been shown that the contents of Spd and Spm in the EC of coffee were significantly higher than those in non-embryogenic calli (NEC), but there was little difference in Put level. The NEC was black in color, and this cell line was unable to proceed with embryogenesis in subsequent experiments. The EC was creamish in color and friable, and the researchers proposed that the accumulation of Spm was necessary for the transformation from a callus to an EC line [24]. The PA content in EC, immature SE and the zygotic embryo of oak was higher than that in mature embryos and germinating embryos [25]. From EC stage to the early stage of embryonic differentiation, the contents of Put, Spd and Spm increased significantly, and then the level of Put decreased until the embryo stage, while the contents of Spd and Spm remained basically the same. From the EC stage to the early stage of embryonic differentiation, the total concentration of PAs increased by 3.6 times in cotton (*Gossypium hirsutum* L.) [26].

Longan (*Dimocarpus longan* Lour.) is an important subtropical fruit in the family *Sapindaceae*. It is grown in more than ten countries. Longan is a source of traditional medicine because of its polyphenol-rich traits in China [27]. The longan SE system is an important SE model system of woody plants, which has great research significance. The PA content in the normal developing ovules of longan was higher than that in the abortive ovules at all stages of embryonic development, and the peak of PA biosynthesis preceded the synthesis of nucleic acid and protein [28]. The study on the SE of longan also showed that the content of endogenous PAs in the EC was higher than that in the NEC. The changes to endogenous PA content always preceded the morphological changes during SE [29]. PAs can also significantly delay the senescence of longan EC, and mostly affect growth and cell viability in its late growth stage [30]. During the development of longan fruit, the change in endogenous PA content was basically consistent with the development of longan fruit, and the PA content also increased rapidly when the pericarp and seed grew rapidly [31]. At present, only a few species have reported the genes related to PA biosynthesis or its metabolism, which suggests that there may be a range of biosynthetic pathway gene families [11,12,16]. However, the gene identification for the whole process of plant PA biosynthesis and metabolism has not been reported. Additionally, the roles of PAs in the critical stage of plant globular embryos (GE) morphogenesis have also been rarely reported upon.

The research into plant SE showed that PAs had a significant interaction with ethylene. Based on the whole-genome identification and bioinformatic analysis of the genes involved in PA biosynthesis and its metabolism family in the longan genome, ACC, D-arginine (D-arg) and PAs were added to a longan GE induction medium to analyze their effects on the transformation of longan EC into GE. Among of them, exogenous D-arg can inhibit the biosynthesis of endogenous PAs, and exogenous ACC can affect the biosynthesis of endogenous PAs by regulating the biosynthesis of endogenous ethylene in plants. The contents of endogenous hormones and H_2_O_2_ and the expression level of PA biosynthesis and its metabolism genes were measured. This study laid a foundation for further determining the physicochemical properties and molecular evolution characteristics of the PA biosynthesis and metabolism gene family members, and exploring the mechanism of ethylene and the PAs regulating the transformation of plant EC into GE.

## 2. Materials and Methods

### 2.1. Plant Materials and Treatments

The experiment was conducted in the Institute of Horticultural Biotechnology at the Fujian Agriculture and Forestry University from 2019 to 2021. The test material was EC of ‘honghezi’ longan, which was induced and subcultured through Zhong-xiong Lai’s methods [32]. Additionally, it was subcultured in our laboratory. Here, 0.01 mmol∙L^−1^ of ACC, D-arg, Put, Spd and Spm were each added to their own GE-inducing medium, and the same volume of distilled water was added to GE-inducing medium as a control. The GE-inducing medium for the EC of longan was MS medium, with 20g∙L^−1^ sucrose and 7g∙L^−1^ agar, and the pH was between 5.8 and 6.0. The test materials were inoculated in four positions of the culture dish, and the total inoculation amount was 0.1 g. After 9, 11, 13 and 15 days of culturing, the treated samples were collected and observed under an optical microscope (DMI8). Then, they were immediately frozen in liquid nitrogen and stored at −80 °C for follow-up tests. Each medium was inoculated with 20 culture dishes, and the experiment was repeated three times.

### 2.2. Identification of Genes Involved in PA Biosynthesis and Metabolism in Longan

PacBio Sequel was used to construct the third-generation genomic high-quality HiC sequencing data for longan, which was stored in our laboratory. Additionally, the second-generation genome data for longan was downloaded from GigaDA (http://gigadb.org/, accessed on 16 March 2021) [27]. The genes involved in PA biosynthesis and its metabolism in *Arabidopsis* were obtained from TAIR (http://www.arabido-psis.org/, accessed on 16 March 2021). They were used as base alignment sequences with the local BLAST of TBtools [33]. All of the sequences identified for PA biosynthesis and its metabolism genes in longan were subjected to the NCBI’s Batch CD-search (https://www.ncbi.nlm.nih.gov/Structure/bwrpsb/bwrpsb.cgi, accessed on 17 March 2021) and SMART (http://smart.embl-heidelberg.de/, accessed on 17 March 2021) tools to verify their reliability as targets [11]. We also examined the physicochemical properties of PA biosynthesis and its metabolism proteins, including determining the number of amino acids (aa), as well as the molecular weight (MW), isoelectric point (pI), instability index (II), aliphatic index (AI) and hydropathicity using ExPASy (https://www.expasy.org/, accessed on 19 March 2021). The subcellular localization of PA biosynthesis and its metabolism proteins were predicted with Plant-PLoc (http://www.csbio.sjtu.edu.cn/bioinf/plant/, accessed on 19 March 2021) [34]. The signal peptide, and the number of transmembrane domain (TMD) and phosphate sites were analyzed with PrediSi (https://www.predisi.de/, accessed on 19 March 2021), TMHMM Server2.0 (https://www.cbs.dtu.dk/services/TMHMM/, accessed on 19 March 2021) and NetPhos3.1Server (http://www.cbs.dtu.dk/services/NetPhos/, accessed on 19 March 2021). The *cis*-acting elements were analyzed with PlantCARE (http://bioinformatics.psb.ugent.be/webtools/plantcare/html/, accessed on 19 March 2021), which were present from 2000 bp upstream of the gene initiation codon. MEME (https://meme-suite.org/meme/tools/meme, accessed on 19 March 2021) was used to analyze the proteins’ conserved motif. Finally, TBtools software (v1.098669, South China Agricultural University, Guangzhou, China) was used to annotate the gene structure and visualize the conserved domains and motifs.

### 2.3. Collinearity Analysis of the Genes Involved in PA Biosynthesis and Metabolism

MCScanX software was used to predict the collinearity of the genes of PA biosynthesis and metabolism, and to mark the collinearity genes, including the collinearity between the longan and *Arabidopsis* species and within the longan species [35].

### 2.4. Phylogenetic Analysis of the Gene Families Involved in PA Biosynthesis and Metabolism

In order to compare the phylogenetic relationship of the genes involved in PA biosynthesis and metabolism in longan with their counterparts in other plant species, we obtained the genes of *Arabidopsis* from TAIR (http://arabidopsis.org, accessed on 16 March 2021), and the rice (*Oryza sativa*) genes from Phytozome v12.1 (https://phytozome.jgi.doe.gov/pz/portal.html, accessed on 16 March 2021). The amino acid sequences of orange (*Citrus sinensis*), apple (*Malus domestica*), cacao (*Theobroma cacao*), cotton (*Gossypium hirsutum*) and potato (*Solanum tuberosum*) were downloaded from NCBI (https://www.ncbi.nlm.nih.gov/, accessed on 16 March 2021). The neighbor-joining method was used to construct the homologous and phylogenetic trees with 1000 bootstrap replicates via MEGA (Molecular Evolutionary Genetics Analyses). Evolview (https://www.evolgenius.info/evolview/#login, accessed on 24 March 2021) was used to visualize the phylogenetic tree and groups [36].

### 2.5. The Analysis of Fpkm Expression Profile of Genes Involved in PA Biosynthesis and Metabolism

Based on the third-generation genome data and the transcriptome data of three biological repeats from different early SEs (EC, incomplete embryonic compact structure (ICpEC) and GE), the fragments per kilobase of exon model per million mapped fragments (FPKM) value of isoform transcripts were analyzed with Excel and visualized with TBtool software [33].

### 2.6. Determination of Endogenous Hormone Levels and H_2_O_2_ Content in Longan Treated with Exoge-Nous ACC, D-arg and PAs

The levels of auxin indole-3-acetic acid (IAA), gibberellin (GA), salicylic acid (SA), abscisic acid (ABA) and hydrogen peroxide (H_2_O_2_) were determined with a double antibody one-step sandwich enzyme-linked immunosorbent assay (ELISA). The specific steps were as follows: add the standard, sample extract and horseradish peroxidase (HRP)-labeled detection antibody to the coating micropores, added with their corresponding antibodies, and incubate and wash thoroughly. The substrate 3,3’,5,5’-tetramethylbenzidine was used for color development. TMB was transformed into blue under the catalysis of peroxidase, and finally into yellow under the action of acid. The color depth was positively correlated with the content of the substance to be measured in the sample. The absorbance was determined with a microplate reader (Infinite M200 PRO, Tecan, Männedorf, Switzerland) at a 450 nm wavelength to calculate the sample activity. The release of ethylene was determined with an ethylene gas detection system (ETD-300, Nijmegen, Netherlands). The required sample was 1.0 g (fresh weight) for the determination of ethylene release. The “stop and flow” mode was adopted for measurement. The accumulation time was 60 min, the determination time was 15 min and the gas flow rate was 3.00 L∙h^−1^.

### 2.7. RNA Extraction and Real-Time Fluorescence Quantitative Polymerase Chain Reaction (qRT-Pcr)

The total RNA of the materials has been extracted using a TransZol Up reagent kit (Transgen, Beijing, China). The RNA quality was analyzed with a Nanodrop 2000 spectrophotometer (Thermo Scientific, Wilmington, DE, USA). The first-strand cDNA was synthesized with a PrimeScript™ IV 1st Strand cDNA Synthesis Mix Kit (TaKaRa, Beijing, China) for qRT-PCR. The DNAMAN software was used to design the primers of the qRT-PCR (Table 1), and the qRT-PCR (Roche Lightcyler 480, Roche, Basel, Switzerland) used was SYBR Green chemistry (2 × SYBR Green PCR Master Mix, Xiamen Biocontrast SciTech Co, Ltd., Xiamen, China). *DlFeSOD* (EU330204) was used as a reference gene for internal control [37]. The operating parameters of the qRT-PCR are as follows: 95 ℃ for 30 s, followed by 40 cycles of 95 ℃ for 10 s and 58 ℃ for 30 s. The relative expression of the genes involved in PA biosynthesis and metabolism were calculated via the 2^− ΔΔCt^ method [37].

## 3. Results

### 3.1. Identification of Genes Involved in PA Biosynthesis and Metabolism

A total of 20 genes involved in PA biosynthesis and metabolism were identified in the longan third-generation genome, including a member of the ADC gene family (DlADC), a member of the ODC gene family (DlODC), three members of the SAMDC gene family (DlSAMDC), two members of the SPDS gene family (DlSPDS), a member of the SPMS gene family (DlSPMS), eight members of the CuAO gene family (DlCuAO) and four members of the PAO gene family (DlPAO). The physicochemical properties of these family proteins are shown in Table 2.

They are encoding proteins with lengths of 328 to 806 aa. The MW of these 20 proteins ranged from 35.93 to 89.80 kDa. The pI ranged from 4.77 to 6.75, meaning that they are acidic proteins. The II were computed to be from 23.31 to 50.54. The proteins of DlSAMDC4, DlCuAO2A, DlCuAO2B, DlCuAO2C, DlCuAO3B, DlPAO2, DlPAO4A and DlPAO4B were classified as stable (II < 40), and the others were classifies as unstable (II > 40). The AI ranged from 77.01 to 99.90. The grand average of hydropathicity (GRAVY) values of all members were negative, meaning that they are hydrophilic proteins. Only six members have signal peptides, namely, DlCuAO2A, DlCuAO2B, DlCuAO2C, DlCuAO3B, DlCuAO and DlPAO5. DlCuAO, DlCuAO2B and DlCuAO3A of the CuAO family have a TMD. DlPAO5 contains the largest number of phosphorylation sites (71); DlSPMS contains the least (28). The main phosphorylation sites of most members were serine (S) sites, followed by threonine (T) sites, and the least were tyrosine (Y) sites. Only DlCuAO2A was predicted to localize in the plasma membrane. DlCuAO3A and DlCuAO3B were predicted to localize in the cell wall. Six members were found primarily in cytoplasm, including DlODC, DlSAMDC4, DlSPDS1A, DlSAMS, DlCuAO2C and DlPAO5. The other 11 members were predicted to localize in the chloroplast.

### 3.2. Chromosomal Localization and Collinearity Analysis of Genes Involved in PA Biosynthesis and Metabolism

A total of 20 genes involved in PA biosynthesis and metabolism were located in eight known chromosomes of longan (Figure 1A). The chromosomal distribution of these genes varied significantly. The largest number of members were located on chromosome1 (Chr1), including *DlSAMDC4*, *DlCuAO2A*, *DlCuAO2B*, *DlCuAO2C*, *DlCuAO3A*, *DlCuAO3B* and *DlPAO5. DlODC* was located in Chr6. *DlSAMDC3*, *DlCuAOB* and *DlSPDS1A* were located in Chr8. *DlSPMS*, *DlPAO4A* and *DlPAO4B* were located in Chr9. *DlADC2* was located in Chr11. *DlPAO2* was located in Chr13. *DlSAMDC2* was located in Chr14. *DlCuAO*, *DlCuAOA* and *DlSPDS1B* were located in Chr15.

When genes of the same family were located in the same chromosome, and the physical distance was more than 70 % within 200 kb, it was defined as a tandem replication event. In the previous research of our group, it was found that 7.59 % of tandem repeats existed in the longan genome [27]. In this research, significant tandem repeat events occurred in the DlCuAO gene family, where *DlCuAO2A*, *DlCuAO2B*, *DlCuAO2C*, *DlCuAO3A* and *DlCuAO3B* were located in the continuous position of Chr1. *DlPAO4A* and *DlPAO4B* of the DlPAO gene family exhibited a tandem repeat event, too. Among the 20 genes in the longan genome, three pairs had a collinearity relationship, including *DlSAMDC2*–*DlSAMDC3*, *DlSPDS1A*–*DlSPDS1B* and *DlPAO4A*–*DlPAO4B*.

The gene families involved in PA biosynthesis and metabolism in the longan and *Arabidopsis* genomes were also analyzed with colinear analysis (Figure 1B). We suggest that 45% of longan genes (9/20) were conserved, compared with *Arabidopsis* genes, and we also suggest that 13 pairs of collinearity genes exist between longan and *Arabidopsis*. Among them, *DlADC2* (Chr11), *DlSAMDC2* (Chr14), *DlSPDS1A* (Chr8) and *DlSPDS1B* (Chr15) corresponded to two genes in *Arabidopsis*. Meanwhile, *AtSPDS1* (Chr1) and *AtSPDS2* (Chr1) also corresponded to two genes in longan.

### 3.3. Phylogenetic Trees, Conserved Motifs and Conserved Domain Analysis of Gene Families Involved in PA Biosynthesis and Metabolism

To gain insight into the evolutionary relationship among the plant genes involved in PA biosynthesis and metabolism, we compared the full length of these protein sequences as found in *Dimocarpus longan* Lour., *Arabidopsis*, *Oryza sativa* L., *Malus domestica*, *Citrus sinensis*, *Theobroma cacao*, *Gossypium hirsutum* and *Solanum tuberosum*. We built the phylogenetic trees, including the ADC-ODC tree, SAMDC tree, SPDS-SPMS tree, CuAO tree and PAO tree (Figure 2).

ADC and ODC belong to group IV pyridoxal-dependent decarboxylases, so we built their phylogenetic tree together (Figure 2A). The ADC-ODC phylogenetic tree was composed of 22 members, including 13 members of ADC family (longan, apple and sweet orange had one member; *Arabidopsis*, rice, cocoa, cotton and potato had two members) and nine members of the ODC family (longan, rice, sweet orange, apple and potato had one member; cocoa and cotton had two members; *Arabidopsis* had no ODC family member). The species that make up the evolutionary tree of the ADC and ODC families were highly conserved in terms of number of members, conserved motifs and domains. Motif 1 and motif 4 were found in the ADC and ODC families, which might be characteristic motifs of the group IV pyridoxal-dependent decarboxylases. All members of the ODC family have the same motifs, among which there are six unique motifs, including motifs 11, 14-17 and 22. Other motifs are unique to the ADC family. However, one or two members of the ADC family are missing motifs 3, 9, 20, 21 and 23. *DlADC* and *DlODC* were close to the ADC and ODC family members of sweet orange.

The phylogenetic tree of the SAMDC family was composed of 31 members (longan, sweet orange and cocoa had three members; *Arabidopsis*, rice and apple had four members; cotton and potato had five members), and it could be divided into SAMDC-Ⅰ, SAMDC-Ⅱ and SAMDC-Ⅲ (Figure 2B). Motifs 1-8 and motif 10 exist in all three subfamilies, and might be the characteristic motifs of SAMDC family. All members of SAMDC-Ⅰ and Ⅱ had motif 9, and most members had motif 11. Motif 14 only appeared in three members of SAMDC-Ⅰ. Motif 15 is the characteristic motif of the SAMDC-Ⅲ members, and motifs 12-13 only appear in the SAMDC-Ⅲ members. The members of the DlSAMDC family were only located in SAMDC-Ⅰ and SAMDC-Ⅲ, which were closed to the SAMDC family members of cotton and sweet orange. *DlSAMDC2* and *DlSAMDC3* had all the unique motifs of this subfamily, but *DlSAMDC4* lacks motif 8 and motif 13. The SAMDC family members contain two kinds of gene structures, one with SAMDC leader peptide (AdoMetDC_leader and AdoMetDC_leader superfamily) and the other with only one conserved domain (PLN02524), and the SAM_Decarbox superfamily also belongs to PLN02524.

SPDS and SPMS were the members of the superfamily cl17173, so we also built their phylogenetic tree together (Figure 2C). The SPDS-SPMS tree was composed of 29 members, including 19 members of the SPDS family (rice had one member; longan, *Arabidopsis*, sweet orange and potato had two members; cocoa and cotton had three members; apple had four members) and ten members of the SPMS family (longan, *Arabidopsis*, sweet orange, cocoa, cotton and potato had one member; rice and apple had two members). The phylogenetic tree of SPDS-SPMS could be divided into SPDS-Ⅰ, SPDS-Ⅱ, SPDS-Ⅲ and SPMS. *DlSPDS1A*, *DlSPDS1B* and *DlSPMS* were located in SPDS-Ⅰ, SPDS-Ⅱ and SPMS, respectively. All members had motifs 1, 2, 5 and 6, which might be the characteristic motifs of the superfamily cl17173. Most of the members had motifs 3, 4, 7, 8, 10 and 11. Motif 13 was the specific motif of SPDS-Ⅰ, and motif 9 and motif 15 were specific to SPMS. *DlSPDS1A* and *DlSPDS1B* had all the characteristic motifs of their subfamilies. *DlSPMS* lacked motifs 3, 4 and 10. The SPDS and SPMS family members contain two kinds of gene structures, which were PLN02366 and the AdoMet_MTases superfamily. The latter belonged to PLN02366. Combining the motif analyses, it was found that *DlSPMS* and *GhSPDS2* within this domain were missing some motifs, which led to their shorter length than PLN02366 in the conservative domain.

The CuAO evolutionary tree was composed of 55 members (rice had four members; sweet orange, apple, cocoa and cotton had six members; longan had eight members; *Arabidopsis* had nine members; potato had ten members, which is the largest in number). Additionally, it could be divided into CuAO-Ⅰ, CuAO-Ⅱ and CuAO-Ⅲ (Figure 2D). *DlCuAO2A*, *DlCuAO2B*, *DlCuAO2C*, *DlCuAO3A* and *DlCuAO3B* were located in CuAO-Ⅰ. *DlCuAO* was located in CuAO-Ⅱ. *DlCuAOA* and *DlCuAOB* were located in CuAO-Ⅲ. All members had motifs 2, 5, 7, 8, 11, 12 and 14, which might be the characteristic motifs of the CuAO family. Motif 15 was unique to CuAO-Ⅲ. The rest of the motifs were distributed across all three subfamilies. In DlCuAO, in addition to *DlCuAO2C* (which was missing motifs 3 and 9), the other members have all the characteristics motifs of their subfamilies. The CuAO family members contain two kinds of gene structures: one was PLN02566, which belonged to CuAO-Ⅰ and CuAO-Ⅱ, and the other was the tynA superfamily, which belonged to CuAO-Ⅲ.

The PAO evolutionary tree was composed of 34 members (cocoa had three members; longan, apple, cotton and potato had four members; *Arabidopsis*, rice and sweet orange had five members). It could be divided into PAO-Ⅰ, PAO-Ⅱ and PAO-Ⅲ (Figure 2E). *DlPAO2*, *DlPAO4A* and *DlPAO4B* were located in PAO-Ⅲ. *DlPAO5* was located in PAO-Ⅱ. All the members had motifs 2, 4-7 and 12, which might be the characteristic motifs of the PAO family. Motif 1 was distributed across all three subfamilies; motifs 3, 8 and 10 are specific motifs of PAO-Ⅰ; motif 9 was distributed across PAO-Ⅰ and PAO-Ⅱ; motifs 11 and 13 were distributed across PAO-Ⅰ and PAO-Ⅲ; motif 15 was distributed across PAO-Ⅱ and PAO-Ⅲ; motif 14 is a characteristic motif of PAO-Ⅲ. In DlPAO, in addition to *DlPAO2* (lacking motif 8), the other members have all the characteristics motifs of their subfamilies. The PAO family members contain three kinds of gene structures. The first was PLN02268, which belonged to PAO-Ⅰ; the second was PLN02568, which belonged to PAO-Ⅱ; the last one was PLN02676, which belonged to PAO-Ⅲ.

### 3.4. Analysis of Structure and Cis-Acting Elements of Genes Involved in PA Biosynthesis and Metabolism in Longan

The *cis*-acting elements of promoters play an important role in regulating gene expression. Different *cis*-acting elements might have an impact on the diverse biological functions of these genes. A total of 16 *cis*-acting elements were predicted to be in the promoters of all the genes involved in PA biosynthesis and metabolism in longan (Figure 3A), including in the phytohormone response elements, the abiotic stress response, meristem expression, zein metabolism, circadian control, flavonoid biosynthesis and the MYB binding site. The DlSPDS-DlSPMS family members responded to the fewest *cis*-acting elements, only eight. There were more than ten types of *cis*-acting elements in the other family members. All of the genes contained a large number of light-responsive elements. (The number of the genes’ light-responsive elements ranged from 8 to 26, with an average of 13.25. In order not to affect the visualization of other *cis*-acting elements, we omitted the light-responsive elements in Figure 3A.) Almost all gene promoters had anaerobic induction elements, except for *DlCuAOα3A*. More than half of the members responded to abscisic acid, auxin, gibberellin, methyl jasmonate (MeJA), salicylic acid (SA) and drought. Moreover, we found that *DlADC2* had a wound responsiveness element, *DlCuAOA* had a flavonoid biosynthetic element, and both *DlSAMDC3* and *DlCuAO3B* responded to circadian control.

The gene structure types of the genes related to PA biosynthesis and metabolism varied in longan (Figure 3B). The first contained only one or more coding sequences (CDS) (*DlADC2*, *DlODC*, the DlSAMDC family members, *DlSPDS1B*, *DlSPMS*, *DlCuAO2B*, *DlCuAO2C*, *DlPAO4A* and *DlPAO4B*). The second type of gene structure has 5′-untranslated regions (5′-UTR) and CDS (*DlCuAOA* and *DlPAO2*). The third gene structure has 3′-UTR and CDS (*DlCuAOα2A*). Additionally, the last gene structure contained 5′-UTR, CDS and 3′-UTR (*DlSPDS1A*, *DlCuAO*, *DlCuAO3A*, *DlCuAO3B*, *DlCuAOB* and *DlPAO5*).

### 3.5. Expression Analysis of PA Biosynthesis and Metabolism Genes during Early SE in Longan Genome

Gene expression patterns might provide important clues for the study of gene function during early SE. A heat map was drawn according to the FPKM values of the genes involved in PA biosynthesis and metabolism from the longan transcriptome database (Figure 4). The expression patterns of the genes during early SE could be divided into four types. The first type was the continuous increase in the gene expression levels, including *DlODC*, *DlSAMDC4*, *DlSPDS1B*, *DlCuAO2A*, *DlCuAOA*, *DlCuAOB*, *DlPAO2*, *DlPAO4A* and *DlPAO4B*. The second type was the continuous decline of the gene expression levels, including *DlADC2, DlSPDS1A*, *DlSPMS*, *DlCuAO*, *DlCuAO2C* and *DlPAO5*. The third type was the lowest expression level of genes in the ICpEC stage, including *DlCuAO3A* and *DlCuAO3B*. The last type was the highest expression level of genes in the ICpEC stage, including *DlSAMDC2*, *DlSAMDC3* and *DlCuAO2B*.

During early SE in longan, the expression level of *DlODC* continuously increased, and the expression level of *DlADC2* continuously declined, but the expression level of *DlADC2* was significantly higher than that of *DlODC*. Therefore, Put was mainly synthesized by *DlADC2* in the early stage of SE in longan, and *DlODC* might play a more and more important role in the middle and late stage of SE. *DlSAMDC2* and *DlSAMDC3* of the DlSAMDC family, *DlSPDS1A* of the DlSPDS family and *DlSPMS* were highly expressed; hence, the synthesis of dcSAM, SPDS and SPMS was mainly caused by these genes during the early SE of longan. In the DlCuAO and DlPAO families, four and three members were highly expressed, respectively. Additionally, the expression level of these genes was increased continuously. These results suggest that the metabolism of PAs might be gradually enhanced during the SE process of longan.

### 3.6. Effects of Exogenous ACC, D-arg and PAs on the Conversion of EC into GE in Longan

The exogenous application of ACC, D-arg and PAs could change the process of transforming EC into GE in longan (Figure 5). The microscopic examination results showed that the period between the 9th and 11th days was key for the transformation of EC into GE in longan. On the 9th day of culture, CK was still in the ICpEC stage, and no GE was found in the visual field. On the 11th day, about half of the calli in the field of vision reached the GE stage. In the observations on the 13th and 15th days, most of the materials had reached the GE stage, and the compactness of the GE was higher. Among the five treatments, 0.01 mmol∙L^−1^ ACC, Put and Spm could promote the transformation of EC into GE, and Spm treatment had the best effect. About half of the materials in the visual field reached the GE stage by the 9th day, and most of them reached the GE stage by the 11th day in the treatment with 0.01mmol∙L^−1^ Spm. The development degree of 0.01 mmol∙L^−1^ Spd treatment was lower than CK on the 9th and 11th days, and was better than CK on the 13th and 15th days. In addition, 0.01 mmol∙L^−1^ D-arg treatment inhibited the transformation of EC into GE. Only a small number of GE were observed on the 11th day, and the number of GE increased on the 13th and 15th days, but this was mainly deformed GE.

### 3.7. Effects of Exogenous ACC, D-arg and PAs on Endogenous Hormone Level and H_2_O_2_ Content in Longan

Since the 9th and 11th days mark the key periods for the morphogenesis of EC into GE, our analysis of endogenous hormone levels and H_2_O_2_ content mainly focuses on these two time points. IAA measurements showed that a low level of IAA was conducive to the morphogenesis of GE. Exogenous ACC, D-arg, Spd and Spm increased the level of endogenous IAA in longan at the later stage of culture (Figure 6A). Higher levels of GA were beneficial to the morphogenesis of GE, and with the further maturation of GE, the GA level decreased. Exogenous D-arg decreased the endogenous GA level, while exogenous ACC increased the endogenous GA level of longan in the whole determination stage (Figure 6B). Ethylene release gradually decreased during longan GE morphogenesis and maturation. Exogenous D-arg and PA treatment mainly reduced ethylene release. Exogenous ACC treatment could increase ethylene release in the early stage of culture (ACC was the precursor of plant ethylene synthesis) (Figure 6C). ABA increased first and then decreased in the determination stage. A higher level of ABA could promote the morphogenesis of GE (Figure 6D). The SA level mainly showed an upward trend during the morphogenesis and maturation of longan GE, and a higher level of SA could promote the morphogenesis of longan GE (Figure 6E). H_2_O_2_ was the main ROS involved in cell signal transduction, and its content gradually decreased during longan GE morphogenesis and maturation. A lower level of H_2_O_2_ was conducive to longan GE morphogenesis (Figure 6F).

### 3.8. Expression Analysis of PA Biosynthesis and Metabolism Family Members under Exogenous ACC, D-arg and PAs Treatment

The addition of exogenous ACC, D-arg and PAs significantly affected the expression of the PA biosynthesis and metabolism gene family members (Figure 7). Here, 0.01 mmol∙L^−1^ ACC treatment significantly upregulated the expression levels of *DlADC2*, *DlODC*, *DlSPDS1A*, *DlCuAOB* and *DlPAO4B*, and significantly downregulated the expression levels of *DlSAMDC2* and *DlSPMS*. In addition, 0.01 mmol∙L^−1^ D-arg treatment significantly increased the expression levels of *DlADC2*, *DlODC* and *DlCuAOB*, and significantly decreased the expression levels of *DlSAMDC2* and *DlSPMS*, but had little effect on the expression levels of *DlSPDS1A* and *DlPAO4B*. Under 0.01 mmol∙L^−1^ Put treatment, the expression patterns of the PA biosynthesis and metabolism family members were divided into three categories: the first was those with significantly upregulated gene expression levels, including *DlADC2*, *DlODC*, *DlSPDS1A*, *DlCuAOB* and *DlPAO4B*; the second is where there was little effect on the gene expression level in the early stage, but where expression levels were significantly promoted in the later stage of culture, and this included only *DlSAMDC2*; the last one is where gene expression levels were significantly inhibited in the early stage, and significantly promoted in the later stage of culture, which included only *DlSPMS*. The gene expression patterns under 0.01 mmol∙L^−1^ Spd treatment were divided into four categories: the first involved significantly upregulated gene expression levels, and included *DlODC* and *DlSPDS1A*; the second involved significantly downregulated gene expression level, and included *DlSAMDC2*, *DlSPMS* and *DlCuAOB*; the third was where gene expression levels were promoted in the early stage, but where there was little effect on gene expression in the later stage of culture, which included only *DlADC2*; the last one is where there was little effect on gene expression levels in the early stage, but where levels were promoted in the later stage of culture, and this included only *DlPAO4B*. The expression patterns of the PA biosynthesis and metabolism family members under 0.01 mmol∙L^−1^ Spm treatment were divided into three categories: the first involved significantly upregulated gene expression levels, and included *DlADC2*, *DlODC*, *DlSPDS1A*, *DlCuAOB* and *DlPAO4B*; the second involved downregulated gene expression levels, and included only *DlSPMS*, the last involved downregulated gene expression levels in the early stage and upregulated levels in the later stage of culture, and included only *DlSAMDC2*.

## 4. Discussion

### 4.1. Structural and Evolutionary Analysis of Members of PA Biosynthesis and Metabolism Family

PAs are a low molecular weight aliphatic nitrogen-containing base with biological activity. They play a regulatory role in many basic cellular and physiological and biochemical processes [38]. The synthesis of PAs in plants involves a variety of enzymes, including ADC, ODC, SAMDC, SPDS and SPMS, while CuAO and PAO constitute the metabolic pathway of PAs [14]. At present, the research on PAs mainly focuses on the most abundant PA levels [14] and the effects of exogenous PAs on the growth and development of animals, plants and microorganisms [39,40,41]. However, there are few reports on the gene families of the PA biosynthesis and metabolism pathway [11,12,15,42]. In this study, based on the third-generation genome data of longan, a total of 20 genes involved in PA biosynthesis and metabolism were found. The DlADC, DlODC and DlSPMS families have only one member; the DlSPDS family has two members; the DlSAMDC family has three members; and the DlCuAO and DlPAO families have eight and four members, respectively. The number of family members were similar to that of *Arabidopsis*, sweet orange and other species, and the phylogenetic analysis showed that the genes of PA biosynthesis and metabolism in longan were closely related with sweet orange. ADC and ODC belong to group IV pyridoxal-dependent decarboxylase, so they have many similarities in gene structure. For example, both the ADC and ODC families have motif 1 and motif 4. ODC was absent in the genomes of some species, such as *Arabidopsis*, so ADC should be an older gene family. In *Arabidopsis*, the *adc* mutant has highly kinked roots that form a tight cluster; it also has narrower leaves, sepals and petals than the wild type, and delayed flowering [43]. It is speculated that *DlADC2* also has the function of regulating the flowering period and the growth of roots and leaves. The SAMDC family was also very conservative. Most members have only one conservative domain, and only a small number of members have lead peptides before the conservative domain. DlSAMDC members have both types of gene structures, but no DlSAMDC family members were found in SAMDC-Ⅱ. All SAMDC family members had motifs 1–8 and motif 10. It was found that when spinach was treated at low temperature, the enhanced activity of SAMDC, with a consequential rise of Spd in chloroplasts, was crucial for the cold acclimation of the photosynthetic apparatus in spinach leaves [44]. We also found *cis*-acting elements in response to low temperatures in *DlSAMDC2*. SPDS and SPMS belonged to the superfamily cl17173. In angiosperms, SPMS evolved from SPDS through gene replication, while *Scots pine*, as a gymnosperm, had a bifunctional SPDS, which could produce Spd and Spm [45]. All members of SPDS and SPMS had motifs 1, 2, 5 and 6. CuAO family members have two conserved domains. PLN02566 belongs to CuAO-Ⅰ and CuAO-Ⅱ, while the TynA superfamily belongs to CuAO-Ⅲ. DlCuAO has two types of conserved domains. In *Arabidopsis*, *AtCuAOδ* (At4g12290) was involved in the H_2_O_2_ production related to ABA-induced stomatal closure [46], which suggests the putative functional regulation of *AtCuAOζ* (AT2g42490) in cellular proliferation and differentiation in the root [47]. In DlCuAO, we found that five members responded to ABA and five members responded to meristem expression. Among them, *DlCuAOA*, *DlCuAO2A* and *DlCuAO2C* had the *cis*-acting elements in response to both ABA and meristem expression. Three different subfamilies of the PAO family have different conserved domains. PLN02268 belongs to PAO-Ⅰ, PLN02568 belongs to PAO-Ⅱ and PLN02676 belongs to PAO-Ⅲ. The DlPAO family lacks the member with PLN02676. In Zhang’s research, mycorrhizas improved the drought tolerance of trifoliate orange by regulating the metabolism of PAs, which was manifested by the higher expression levels of *PtPAO1*, *PtPAO2* and *PtPAO3* in the trifoliate orange inoculated with mycorrhizas, which was not the case in that which was not inoculated [16]. Among the four members of the DlPAO family, we found *cis*-acting elements in response to drought stress. It is suggested that the high expression of DlPAO family members could improve the drought tolerance of longan.

### 4.2. Exogenous ACC, D-arg and PAs Regulate the Morphogenesis of Longan GE by Changing the Level of Endogenous Hormones

Previous studies have shown that treatment with the same concentration of different PA compounds may have either positive or negative effects, depending on the PA applied (Put, Spd or Spm), on the mode of PA treatment, on the investigated plant species and on the stress factor, suggesting that “the more PA the better” is not always true [48,49,50,51,52]. Our research results showed that the period between the 9th and 11th days was key for the transformation of EC into GE, in which Put and Spm treatment promoted the transformation, while Spd treatment inhibited it in the early stage, and D-arg, as an inhibitor of PA biosynthesis, also inhibited this process. Exogenous Spd treatment could reduce the levels of endogenous IAA, GA and ABA in rice seedlings [53]. The study on germinating maize seeds showed that the application of Spd significantly increased the contents of endogenous Spd, GA, ethylene and H_2_O_2_, but decreased the concentration of ABA in embryos [54]. The regulation of PAs in strawberry fruit ripening was also closely related to the levels of ABA, ethylene and IAA, and the up- or downregulation of strawberry SAMDC expression levels promotes or inhibits the accumulation of ABA, respectively, and inhibits or promotes the accumulation of IAA, respectively, but both inhibit the release of ethylene [7]. The external application of Spm inhibited ABA synthesis and improved plant reproductive health in osmotic-stressed soybean pods [21]. In our study, more than 50% of the PA genes responded to ABA, auxin, GA, MeJA and SA. Exogenous ACC, D-arg and PA treatments significantly affected the level of endogenous hormones and the content of H_2_O_2_ in longan. During the critical period of longan EC transformation into GE, Put and Spm, which promote the morphogenesis of longan GE, and D-arg, which inhibits the morphogenesis of longan GE, promote the accumulation of SA and inhibit the accumulation of IAA, GA, ethylene and H_2_O_2_. Their role may be to maintain the morphogenesis of longan GE. As a hormone closely related to the stress response, ABA may play a major role in the process of GE morphogenesis in longan, which is manifested in the increase in ABA accumulation in the treatment for promoting GE morphogenesis, while there was a decrease in the ABA level in the D-arg treatment. (Figure 6). Jin and Karami also found that stress, as well as plant hormones, play an important role during plant SE [55,56].

### 4.3. Exogenous ACC, D-arg and PAs Were Involved in Regulating the Expression of PA Biosynthesis and Metabolic Family Members in Longan

The contents of Spd and Spm in the EC of coffee were significantly higher than those in NEC, and the accumulation of Spm was the necessary condition for the transformation of callus to an EC line [24]. The accumulation of PAs was also needed before the differentiation of longan EC, and the change in the endogenous PA content always precedes the change in morphology during SE [29]. In the normal developing ovules of longan, the content of PAs was higher than that of abortive ovules at all stages of embryonic development, and the peak of PA biosynthesis was earlier than that of the nucleic acid and proteins [28]. During the transformation of EC into the early phase of embryo differentiation, the genes related to PA synthesis were mainly downregulated in cotton [26]. The transcriptome data at different early stages of longan SE showed that the *DlADC2*, *DlSAMDC2*, *DlSAMDC3*, *DlSPDS1A* and *DlSPMS* genes involved in PA synthesis were highly expressed in the EC and ICpEC stages, but decreased in the GE stage, which was basically consistent with the results of previous studies. The expression of *DlODC* increased continuously in this process, but its expression was only 1.19% of *DlADC2*, which contributed little to Put synthesis. Exogenous D-arg could significantly reduce the level of endogenous PAs, while exogenous PAs can significantly increase the level of endogenous PAs in cotton, and the kind of PAs added is always the one which leads to the greatest increase in endogenous PAs [26]. Exogenous ACC, D-arg and PAs all changed the expression level of the PA bioanabolic family members in longan. Additionally, the expression levels of *DlADC2*, *DlODC*, *DlSPDS1A*, *DlCuAOB* and *DlPAO4B* were mainly upregulated, while DlSAMDC2 and DlSPMS were mainly downregulated. Interestingly, exogenous Spm treatment always downregulated the expression level of *DlSPMS*, and its relative expression was only 7.60-16.22% of CK. It might be that the exogenous Spm in the medium could be directly absorbed and utilized by EC, and the absorbed Spm was enough for longan, and it thus reduced the synthesis of endogenous Spm (Figure 8).

## 5. Conclusions

In the third-generation genome of longan, a total of 20 genes involved in PA biosynthesis and metabolism were identified. There is only one member in the DlADC, DlODC and DlSPMS families, and there are three, two, eight and four members in the DlSAMDC, DlSPDS, DlCuAO and DlPAO families, respectively. The family members were highly conservative, and have a large number of *cis*-acting elements that respond to light, plant hormones and environmental stress. The period between the 9th and 11th days was key for the transformation of EC into GE in longan. There were higher levels of GA, SA and ABA and lower levels of IAA, ethylene and H_2_O_2_ in the key period. The expression levels in this period of *DlADC2*, *DlODC*, *DlSPDS1A*, *DlCuAOB* and *DlPAO4B* were upregulated, while those of *DlSAMDC2* and *DlSPMS* were downregulated. Exogenous ACC, D-arg and PAs regulated the morphogenesis of longan GE by changing the level of endogenous hormones and the expression level of the PA biosynthesis and metabolism family members. Among them, 0.01 mmol∙L^−1^ of ACC, Put and Spm could promote the transformation of EC into GE, and Spm treatment had the best effect, while the treatment of 0.01 mmol∙L^−1^ D-arg inhibited this process. This study provided a foundation for further determining the physical and chemical properties and molecular evolution characteristics of the PA biosynthesis and metabolism gene family members, and explored the mechanism of ethylene and the PAs regulating the transformation of plant EC into GE.

## Figures and Tables

**Figure 1 genes-13-00317-f001:**
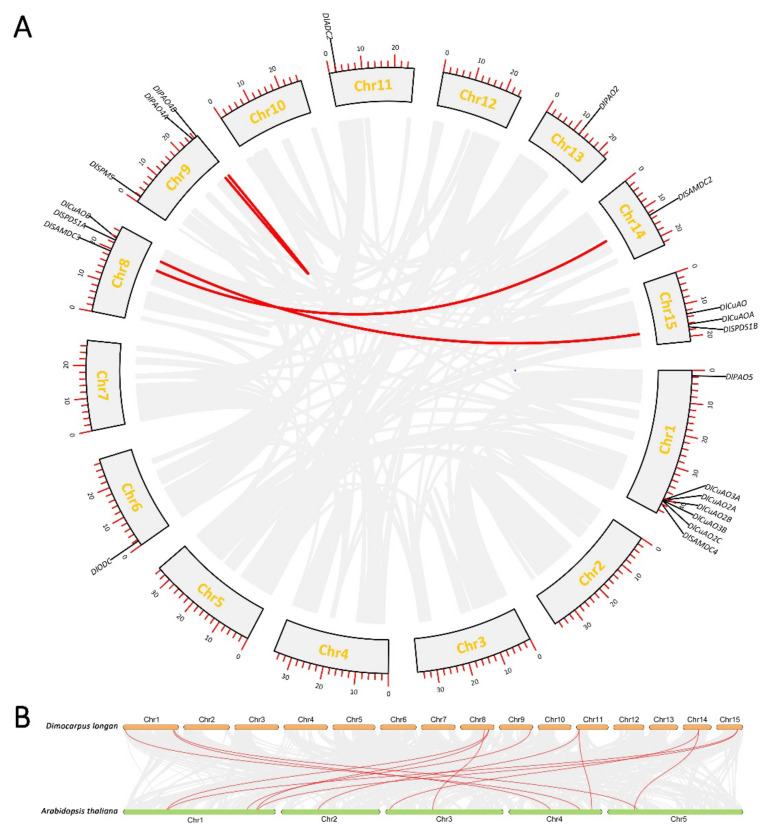
Collinear analysis of genes involved in PA biosynthesis and metabolism. (**A**): Collinearity analysis within longan species. Gray lines indicate the collinear blocks within the longan genome, the red lines represent the syntenic gene pairs relating to PA biosynthesis and metabolism. (**B**): Colinear analysis between longan and *Arabidopsis*. Gray lines indicate the collinear blocks within the longan and *Arabidopsis*, and the red lines represent the syntenic gene pairs relating to PA biosynthesis and metabolism.

**Figure 2 genes-13-00317-f002:**
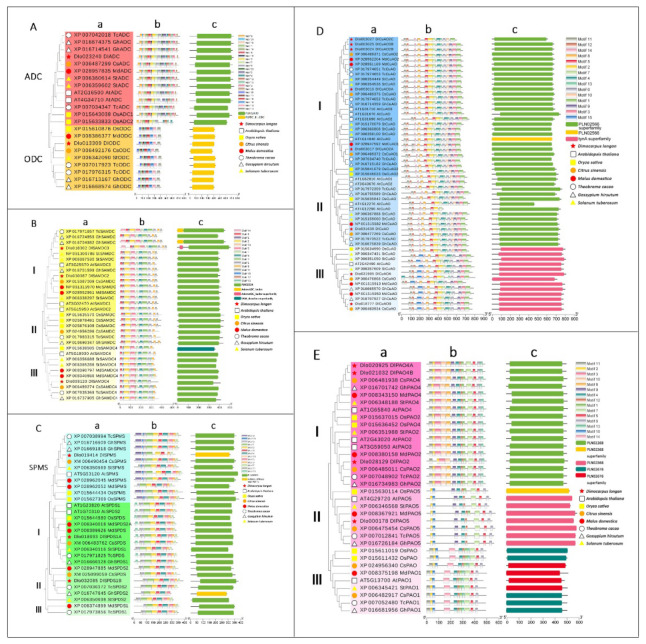
The analysis of the phylogenetic trees, conserved motifs and conserved domains of the PA biosynthesis and metabolism gene family. (**A**): ADC-ODC families, (**B**): SAMDC family, (**C**): SPDS-SPMS families, (**D**): CuAO family, (**E**): PAO family. a: phylogenetic tree, b: gene conserved motifs, c: gene conserved domains.

**Figure 3 genes-13-00317-f003:**
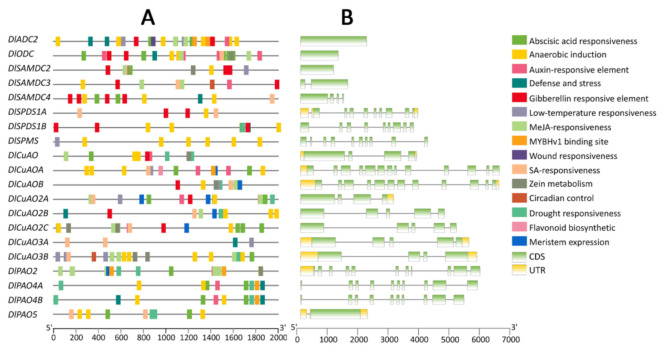
Promoter of *cis*-acting elements and gene structure of genes involved in PA biosynthesis and metabolism of longan. (**A**): *cis*-acting element of genes promoter, (**B**): genes structure.

**Figure 4 genes-13-00317-f004:**
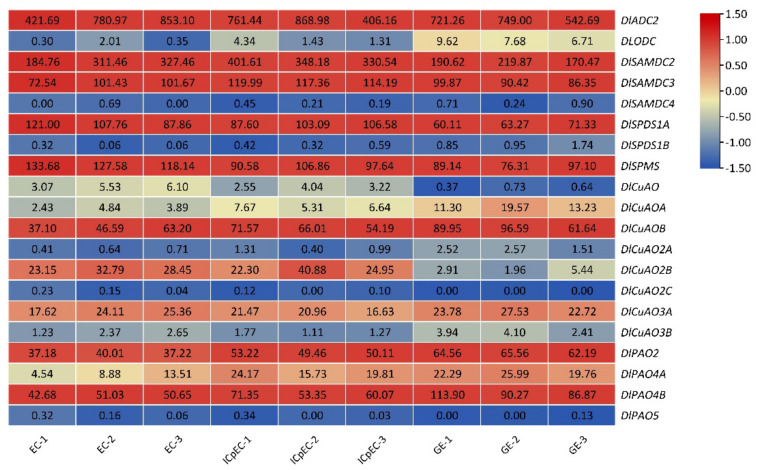
The FPKM of the genes involved in PA biosynthesis and metabolism in different early SE stages of longan.

**Figure 5 genes-13-00317-f005:**
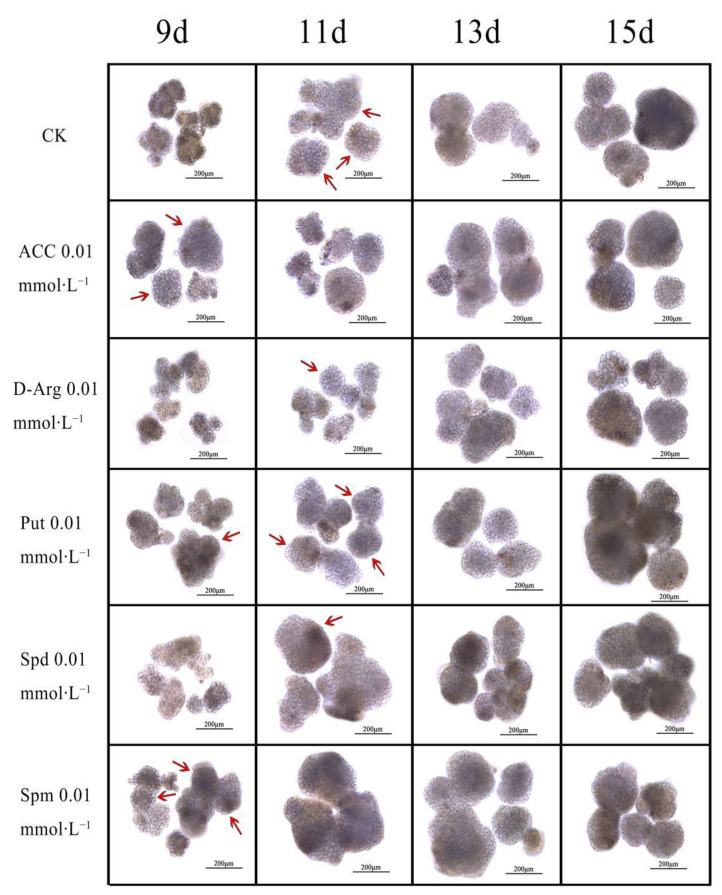
Effects of exogenous ACC, D-arg and PAs on the conversion of EC into GE in longan. Additionally, the first appearance of GE in each treatment is marked by red arrows.

**Figure 6 genes-13-00317-f006:**
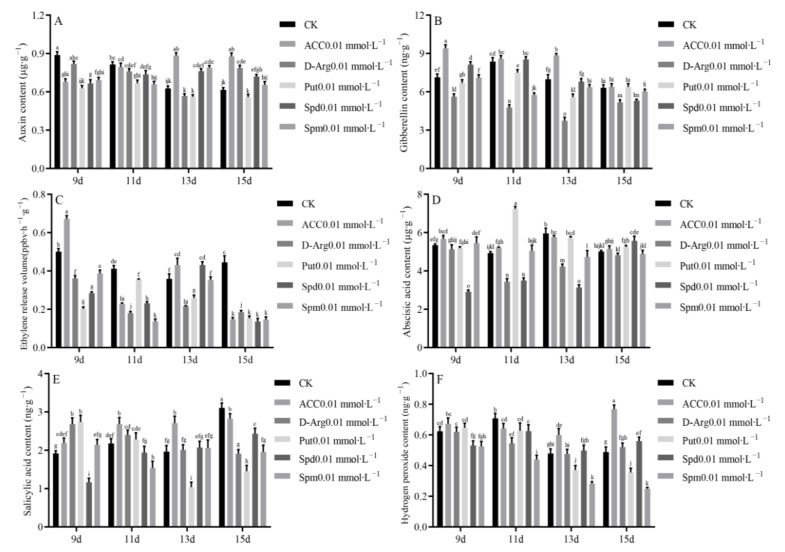
Effects of exogenous ACC, D-arg and PAs on endogenous hormone level and H2O2 Content in longan. (**A**): endogenous IAA content, (**B**): endogenous GA content, (**C**): ethylene release volume, (**D**): endogenous ABA content, (**E**): endogenous SA content, (**F**): endogenous H_2_O_2_ content.

**Figure 7 genes-13-00317-f007:**
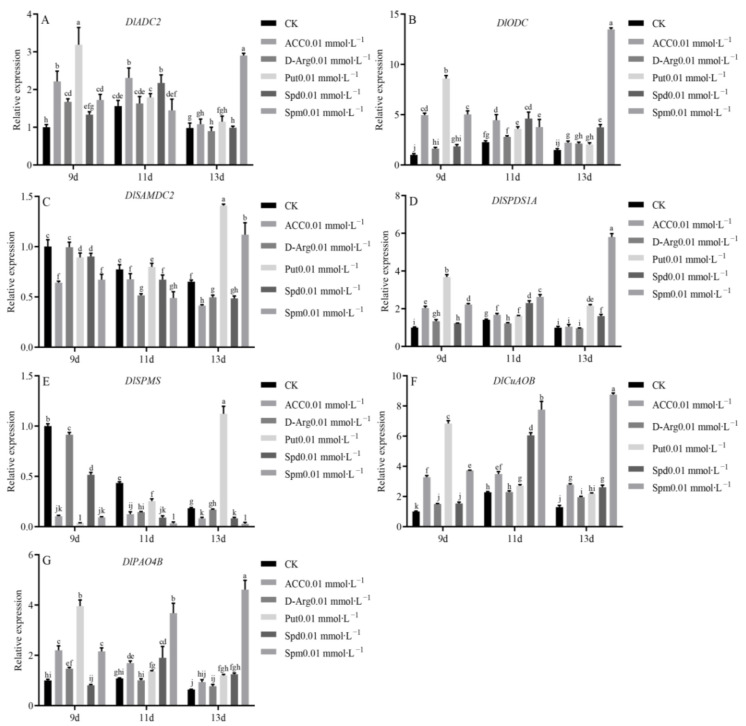
Expression level of the genes involved in PA biosynthesis and metabolism during GE in longan. The equation 2^−ΔΔCt^ was applied to calculate the relative expression level using *DlFeSOD* as the reference gene. (**A**): the relative expression of *DlADC2*; (**B**): the relative expression of *DlODC*; (**C**): the relative expression of *DlSAMDC2*; (**D**): the relative expression of *DlSPDS1A*; (**E**): the relative expression of *DlSPMS*; (**F**): the relative expression of *DlCuAOB*; (**G**): the relative expression of *DlPAO4B*.

**Figure 8 genes-13-00317-f008:**
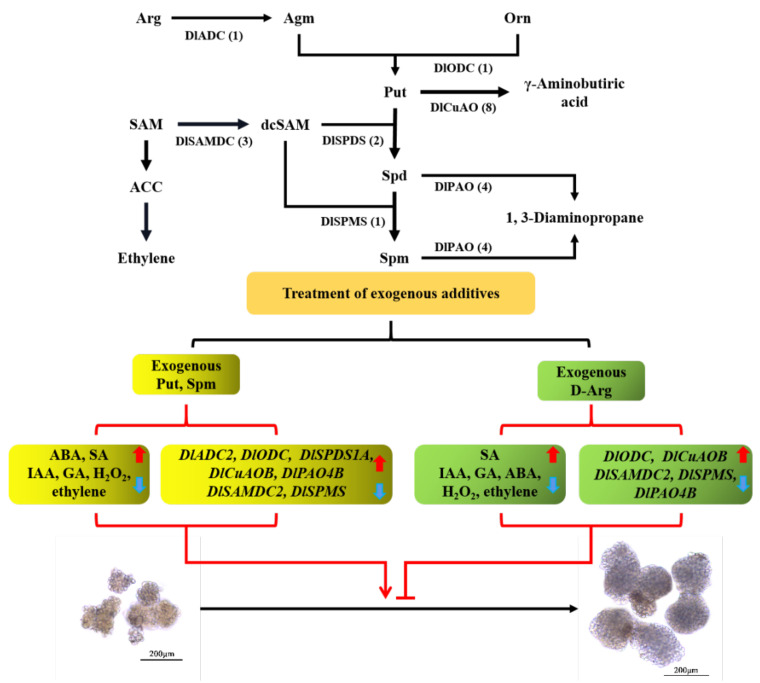
Schematic diagram of changes in the content of plant endogenous hormones and the expression of PA synthesis and metabolism genes during the critical stage of longan GE morphogenesis. The black arrow indicates the basic change process; the number in black brackets is the number of family members; the types of exogenous treatments are in the black curly brackets; the changes to the subjects after treatment are shown in red curly brackets; the upward red mark indicates the increase in endogenous hormone content or the upregulation of gene expression levels, and the downward blue mark indicates the decrease in endogenous hormone content or the downregulation of gene expression levels; the red arrow indicates promoting the process, and the red “T” line indicates inhibiting the process.

**Table 1 genes-13-00317-t001:** Primer sequences and produce size in qRT-PCR analysis.

Gene Name	Primer Sequences (5′ to 3′)		Produce Size (bp)
*DlADC2*	F: CCAGACCCACAACTACGAC	R: CAACCCGAAACGGAAAC	115
*DLODC*	F: GAGTGGCACCCGAAATG	R: CTTGAGCCGAGATGCTTG	143
*DlSAMDC2*	F: CGTTACACCCGAAGATGG	R: GGAGCAAGCAAGAACCC	106
*DlSPDS1A*	F: CGCAGAGAGGGATGAATG	R: GCCGCCAATAACCAGAAC	95
*DlSPMS*	F: CGAATGCCACTCCACTG	R: CGTGTGCTTCTCCAGGC	113
*DlCuAOB*	F: CTTCTTCCCTCCATTCCAG	R: CACAACAAGTCTCGCACG	97
*DlPAO4B*	F: GTTGATACGCTCCCTTGG	R: GGAACTTTGTTGCCGTCC	114
*DlFeSOD*	F: CAGATGGTGAAGCCGTAGAG	R: TATGCCACCGATACAACAAAC	106

**Table 2 genes-13-00317-t002:** Physicochemical properties of the proteins involved in PA biosynthesis and metabolism.

Gene Name	Second-Generation Genome ID	Third-Generation Genome ID	Annotated in *Arabidopsis*	Protein Length (aa)	MW (Da)	pI	II	AI	GRAVY	Signal Peptide	TMD	Phosphorylation Site				Subcellular Localization
												S	Y	T	Total	
*DlADC2*	Dlo_024362.1	Dlo023240	*AtADC2*	726	77813.0	5.23	41.67	90.32	−0.043	No	0	43	11	10	64	chloroplast
*DlODC*	Dlo_001547.1	Dlo013309	*AtODC*	414	44583.8	5.94	40.39	87.22	−0.007	No	0	16	4	11	31	cytoplasm
*DlSAMDC2*	Dlo_037470.1	Dlo030087	*AtSAMDC2*	362	40105.6	4.77	43.64	83.23	−0.083	No	0	26	7	7	40	chloroplast
*DlSAMDC3*	Dlo_013246.1	Dlo018302	*AtSAMDC3*	451	50084.2	5.27	49.47	85.57	−0.057	No	0	43	3	7	53	chloroplast
*DlSAMDC4*	Dlo_029928.1	Dlo003120	*AtSAMDC4*	368	40994.6	5.00	36.36	77.01	−0.179	No	0	23	9	16	48	cytoplasm
*DlSPDS1A*	Dlo_018546.3	Dlo018693	*AtSPDS1*	349	38036.2	4.90	42.10	82.87	−0.136	No	0	15	3	10	28	cytoplasm
*DlSPDS1B*	Dlo_012336.1	Dlo032085	*AtSPDS1*	390	43624.7	5.74	42.89	85.51	−0.171	No	0	28	6	10	44	chloroplast
*DlSPMS*	Dlo_032255.1	Dlo019414	*AtSPMS*	328	35931.3	5.83	50.54	88.48	−0.143	No	0	20	5	3	28	cytoplasm
*DlCuAO*	Dlo_005522.1	Dlo031638	*AtCuAO*	738	83634.3	6.14	40.72	80.41	−0.348	Yes	1	31	9	25	65	chloroplast
*DlCuAOA*	Dlo_019230.1	Dlo031995	*AtCuAO*	806	89799.7	6.62	45.24	80.87	−0.276	No	0	45	7	17	69	chloroplast
*DlCuAOB*	Dlo_019230.1	Dlo018777	*AtCuAO*	774	86787.0	6.69	45.31	78.18	−0.372	No	0	29	4	19	52	chloroplast
*DlCuAO2A*	Dlo_030965.1	Dlo003017	*AtCuAO2*	654	74021.1	6.75	36.00	84.02	−0.247	Yes	0	29	9	25	63	Plasma membrane
*DlCuAO2B*	Dlo_030970.1	Dlo003024	*AtCuAO2*	670	76074.0	6.15	34.77	83.31	−0.287	Yes	1	25	7	22	54	chloroplast
*DlCuAO2C*	Dlo_030974.1	Dlo003027	*AtCuAO2*	603	68096.8	5.97	34.38	86.29	−0.254	Yes	0	25	4	24	53	cytoplasm
*DlCuAO3A*	Dlo_030964.1	Dlo003016	*AtCuAO3*	672	76865.4	6.09	41.18	81.74	−0.336	No	1	30	11	20	61	cell wall
*DlCuAO3B*	Dlo_030971.1	Dlo003025	*AtCuAO3*	670	75847.6	6.49	30.81	83.90	−0.288	Yes	0	25	8	27	60	cell wall
*DlPAO2*	Dlo_000974.2	Dlo028129	*AtPAO2*	498	55240.3	5.63	35.16	94.74	−0.031	No	0	24	3	13	40	chloroplast
*DlPAO4A*	Dlo_010639.1	Dlo020925	*AtPAO4*	486	53906.1	5.98	39.26	99.90	−0.010	No	0	21	4	6	31	chloroplast
*DlPAO4B*	Dlo_010639.1	Dlo021032	*AtPAO4*	486	53828.9	5.93	39.60	99.69	−0.012	No	0	21	4	6	31	chloroplast
*DlPAO5*	Dlo_030297.1	Dlo000178	*AtPAO5*	546	60561.7	5.89	47.15	80.20	−0.295	Yes	0	45	8	18	71	cytoplasm

## Data Availability

Not applicable.

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
