# Peer review of "PAs Regulate Early Somatic Embryo Development by Changing the Gene Expression Level and the Hormonal Balance in Dimocarpus longan Lour."

_genes, 2022, doi:10.3390/genes13020317_

Round 1

Reviewer 1 Report

genes-1533069[1]

Brief Summary:

The manuscript “Identification of genes related to polyamines synthesis and metabolism and the mechanism analysis of regulating globular embryo morphogenesis in response to exogenous polyamines treatments in Dimocarpus longan Lour) is a study of somatic embryo (SE) development in response to polyamines (PAs) and their biosynthesis precursors in longan. They used PacBio data for identification of the genes in PAs metabolic pathway and performed subsequent expression profiling of the target gene, in response to PAs and their precursors, by qPCR. They also performed hormonal analysis to investigate the changes in hormonal balance in response to PAs and their precursors. Their results showed that 1-aminocyclopropane-1-carboxylic acid (ACC), putrescine (Put) and spermine (Spm) can positively regulate SE development, whereas D-arginine (D-Arg) treatment inhibits the process. They finally conclude that changes in gene expression as well as hormonal balance underlie the SE development in a PA dependent manner.

Broad comments:

The study, in its scale, provides interesting data about the PAs control of SE development and is designed and performed in a proper way. The experiments and analyses were also performed carefully with fair number of replicates. The writing and arrangement of the manuscript may however be improved. The discussion text is sometimes the exact repetition of what was reported in the results section and lacks a proper explanation and insights from the authors (refer to the examples below).

Specific comments:

 The title should be changed; it is passive, sounds a bit long with an improper flow. It would be better to have a title with a focus on the action for example: “PAs regulate the SE development through changing the hormonal balance” or sth like this.

Line 43: “can be bound” or “can bind” instead of “can be bind”

Line 58: “PAs” instead of “PAS” I suppose.

Lines 61 and 67: “synthases” instead of “syntheses”

Line 70: “maintains” instead of “works to maintenance”

Line 80: “PAs play” instead of “PAs plays”

Line 82: “has been shown” instead of “has been showed”

Lines 114-115: It would be better to explain the reasoning behind the ACC and D-arginine treatments. It might sound clear to the authors but would help the readers.

Line 177: “genome date” to “genome data”.

Line 226: “were classifies” to “were classified”

Line 243: “in Chr1” to “on chromosome1 (Chr1)”.

Lines 371-372: Better to finish the sentence like: “The gene structure types of the genes related to PAs biosynthesis and metabolism varied in longan”

Line 372:  Better to start with “The first contained”

Line 374: Better to say “the second type” instead of “the second kind”

Figure 3 is missing!!

Line 402: “IAA measurements showed” instead of “The determination results of IAA”

Lines 461-463: This sentence should be rewritten.

Line 467: Should be “In longan were closely related with sweet orange.” I suppose.

Line 503: Should be “by changing”

Line 519-531: This part of discussion is the exact repetition of the results section and lack the insight and the interpretation of the authors. It is clear what hormone levels were changed in response to the PAs but what is really needed here is to explain and even briefly speculate how mechanistically hormonal changes can affect SE development.

Line 539-540: Should be “precedes the change in morphology”

Line 593: “’by changed” to “by changing”

Author Response

Response to Reviewer 1 Comments

Dear Reviewer,

Thank you for your careful review of the manuscript and put forward many valuable opinions, which is of great help to us to further revise the article. Compared with the review report, we made the following modifications to the manuscript.

Point 1: The title should be changed; it is passive, sounds a bit long with an improper flow. It would be better to have a title with a focus on the action for example: “PAs regulate the SE development through changing the hormonal balance” or sth like this.

Response 1: We changed the title of the paper to “PAs regulate the early SE development through changing the genes expression level and the hormonal balance in Dimocarpus longan Lour.”(Line 2-4)

Point 2: Line 43: “can be bound” or “can bind” instead of “can be bind”.

Response 2: In the paper, "can be bind" has been changed to "can be bound" and highlight. (Line 42)

Point 3: Line 58: “PAs” instead of “PAS” I suppose.

Response 3: In the paper, "PAS" has been changed to "PAs" and highlight. (Line 58)

Point 4: Lines 61 and 67: “synthases” instead of “syntheses”.

Response 4: In the paper, "syntheses" has been changed to "synthases" and highlight. (Lines 61, 62 and 67)

Point 5: Line 70: “maintains” instead of “works to maintenance”.

Response 5: In the paper, "works to maintenance" has been changed to "maintains" and highlight. (Line 70)

Point 6: Line 80: “PAs play” instead of “PAs plays”.

Response 6: In the paper, "PAs plays" has been changed to "PAs play" and highlight. (Line 81)

Point 7: Line 82: “has been shown” instead of “has been showed”.

Response 7: In the paper, "has been showed" has been changed to "has been shown" and highlight. (Line 83)

Point 8: Lines 114-115: It would be better to explain the reasoning behind the ACC and D-arginine treatments. It might sound clear to the authors but would help the readers.

Response 8: We provide the reasons behind the ACC and D-arginine treatments in the paper. (Lines 118-120)

Point 9: Line 177: “genome date” to “genome data”.

Response 9: In the paper, "genome data" has been changed to "genome date" and highlight. (Line 183)

Point 10: Line 226: “were classifies” to “were classified”.

Response 10: In the paper, "were classifies" has been changed to "were classified" and highlight. (Line 232)

Point 11: Line 243: “in Chr1” to “on chromosome1 (Chr1)”..

Response 11: In the paper, "in Chr1" has been changed to "on chromosome1 (Chr1)" and highlight. (Lines 250-251)

Point 12: Lines 371-372: Better to finish the sentence like: “The gene structure types of the genes related to PAs biosynthesis and metabolism varied in longan”.

Response 12: In the paper, this sentence has been changed to ”The gene structure types of the genes related to PAs biosynthesis and metabolism varied in longan.” (Lines 383-384)

Point 13: Line 372:  Better to start with “The first contained”.

Response 13: In the paper, "The first was contained" has been changed to "The first contained" and highlight. (Line 384)

Point 14: Line 374: Better to say “the second type” instead of “the second kind”.

Response 14: In the paper, "the second kind" has been changed to "the second type" and highlight. (Line 386)

Point 15: Figure 3 is missing!!

Response 15: We added Figure 3. (Line 390-392)

Point 16: Line 402: “IAA measurements showed” instead of “The determination results of IAA”.

Response 16: In the paper, "The determination results of IAA" has been changed to "IAA measurements showed" and highlight. (Line 439)

Point 17: Lines 461-463: This sentence should be rewritten.

Response 17: We have rewritten this sentence as “The DlADC, DlODC and DlSPMS families have only one member; the DlSPDS family has two members; the DlSAMDC family has three members; and the DlCuAO and DlPAO families have eight and four members, respectively.”. (Lines 502-505)

Point 18: Line 467: Should be “In longan were closely related with sweet orange.” I suppose.

Response 18: We accepted your suggestion and highlight in the paper. (Lines 507-508)

Point 19: Line 503: Should be “by changing”.

Response 19: In the paper, "by changed" has been changed to "by changing" and highlight. (Line 543)

Point 20: Line 519-531: This part of discussion is the exact repetition of the results section and lack the insight and the interpretation of the authors. It is clear what hormone levels were changed in response to the PAs but what is really needed here is to explain and even briefly speculate how mechanistically hormonal changes can affect SE development.

Response 20: We have rewritten this part. You can see the revised manuscript for details. (Lines 565-573)

Point 21: Line 539-540: Should be “precedes the change in morphology”.

Response 21: In the paper, "precedes the changed of morphology" has been changed to "precedes the change in morphology" and highlight. (Line 580)

Point 22: Line 593: “’by changed” to “by changing”.

Response 22: In the paper, "by changed" has been changed to "by changing" and highlight. (Line 630)

We also made some changes to the paper. Please see the revised manuscript for details.

Looking forward to your reply.

Best regards,

Chunwang Lai

Institute of Horticultural Biotechnology, Fujian Agriculture and Forestry University, Fuzhou, 350002

E-mail: laichunwang@163.com

Reviewer 2 Report

The authors describe the role of polyamines on the development from the embryogenic callus to globular embryos. Also, the structures, domains and properties of the proteins analyzed are described in detail. The impact of selected substances on embryogenesis was analyzed and the levels of selected plant hormones were measured.

I recommend proofreading by a native English speaker.

GC to EC: why did D-Arg inhibit the transition? What´s the mechanism? Did you try it with other D-amino acids as well? What other D-amino acids have been described? Do they have a comparable effect? Please elaborate your motivation to use D-Arg a little closer.

In the introduction you describe several biosyntheses. Please give additional information on the interactions of the hormones and precursors and their role in PA activation inhibition. Are there combinatorial effects already described? Did you also test different combinations? If not, you may try this as well.

Table 1: separate the columns and rows with lines, try a smaller font size for the primers that one primer fits in one line, or you could use two separate lines for F and R primers

Figure 2: This figure has a lot of interesting information according to the legend. Please enlarge the different sub figures. I tried to identify it on my computer enlarged to 200% but still could not read it. Put A, B, etc below each other and enlarge the figure to the width of the page. It is big enough when you print it out and can read it easily.

You are referring to Figure 3 which is not in the manuscript!

Figure 4 is not mentioned in the text

Figure 5: please also enlarge this figure

Figure 6+7: Please enlarge these figures as well and increase the font size

14: member

24: 9th and 11th day

61: under the catalysis of spermidine synthase

62: SPM synthase

82: It has been shown

113-114: the sentence is not complete

135-136: was repeated three times

147: including determining the number of…

221: …proteins are shown in Table…

222: Sentence is not complete. Maybe you mean: They are encoding…?

259: we suggest…

260-261: …and we also suggest…

287-288: …were found in…

324: …had four members

346-348: …which belonged to…

381: Exogenous application …

383: …9th and 11th day…

386 + 392 + 395 + 400: …day…

454: PAs are…

458: …the research on…

587: 9th and 11th day…

Author Response

Response to Reviewer 2 Comments

Dear Reviewer,

Thank you for your careful review of the manuscript and put forward many valuable opinions, which is of great help to us to further revise the article. Compared with the review report, we made the following modifications to the manuscript.

Point 1: I recommend proofreading by a native English speaker.

Response 1: We accepted your suggestion and proofread the manuscript many times.

Point 2: GC to EC: why did D-Arg inhibit the transition? What´s the mechanism? Did you try it with other D-amino acids as well? What other D-amino acids have been described? Do they have a comparable effect? Please elaborate your motivation to use D-Arg a little closer.

Response 2: D-Arg is a specific inhibitor of endogenous PAs biosynthesis in plants. (Lines 118-119). Therefore, for the integrity of the test design, we used the D-Arg treatment.

Point 3: In the introduction you describe several biosyntheses. Please give additional information on the interactions of the hormones and precursors and their role in PA activation inhibition. Are there combinatorial effects already described? Did you also test different combinations? If not, you may try this as well.

Response 3: At present, there are many researches on the interactions of the hormones and precursors and their role in PAs activation inhibition. We also introduced it in the discussion part of the manuscript. (Lines 551-560, 570-572).

Point 4: Table 1: separate the columns and rows with lines, try a smaller font size for the primers that one primer fits in one line, or you could use two separate lines for F and R primers.

Response 4: We adjusted the Table 1 and Table 2 to make them more suitable for the typesetting of the paper.

Point 5: Figure 2: This figure has a lot of interesting information according to the legend. Please enlarge the different sub figures. I tried to identify it on my computer enlarged to 200% but still could not read it. Put A, B, etc below each other and enlarge the figure to the width of the page. It is big enough when you print it out and can read it easily.

Response 5: I'm sorry for the inconvenience caused to your review. We have communicated with the editor. we will put all the pictures in a separate folder and provide the pictures with higher definition.

Point 6: You are referring to Figure 3 which is not in the manuscript!

Response 6: We have supplemented Figure 3 in the manuscript. (Lines 391-392)

Point 7: Figure 4 is not mentioned in the text.

Response 7: We have supplemented relevant contents to the manuscript. (Lines 393-414)

Point 8: Figure 5: please also enlarge this figure. Figure 6+7: Please enlarge these figures as well and increase the font size.

Response 8: We have communicated with the editor. we will put all the pictures in a separate folder and provide the pictures with higher definition.

Point 9: 14: member.

Response 9: We revised the manuscript and checked the full text. (Line 13)

Point 10: 24: 9th and 11th day”.

Response 10: We revised the manuscript and checked the full text. (Lines 23, 420, 429, 437, 623)

Point 11: 61: under the catalysis of spermidine synthase. 62: SPM synthase.

Response 11: We revised the manuscript and checked the full text. (Lines 61, 62, 67)

Point 12: 82: It has been shown.

Response 12: We revised the manuscript and checked the full text. (Line 83)

Point 13: 113-114: the sentence is not complete.

Response 13: We modified the sentence. Please see the revised manuscript for details. (Lines 113-114)

Point 14: 135-136: was repeated three times.

Response 14: We revised the manuscript and checked the full text. (Line 140)

Point 15: 147: including determining the number of…

Response 15: We revised the manuscript and checked the full text. (Lines 151-152)

Point 16: 221: …proteins are shown in Table…

Response 16: We revised the manuscript and checked the full text. (Line 227)

Point 17: 222: Sentence is not complete. Maybe you mean: They are encoding…?

Response 17: We modified the sentence. Please see the revised manuscript for details. (Line 228)

Point 18: 259: we suggest… 260-261: …and we also suggest…

Response 18: We revised the manuscript and checked the full text. (Lines 267-269)

Point 19: 287-288: …were found in….

Response 19: We revised the manuscript and checked the full text. (Line 296)

Point 20: 324: …had four members.

Response 20: We revised the manuscript and checked the full text. (Line 334)

Point 21: 346-348: …which belonged to…

Response 21: We revised the manuscript and checked the full text. (Lines 344, 345, 357, 358)

Point 22: 381: Exogenous application ….

Response 22: We revised the manuscript and checked the full text. (Line 417)

Point 23: 381: 454: PAs are…

Response 23: We revised the manuscript and checked the full text. (Line 492)

Point 24: 458: …the research on…

Response 24: We revised the manuscript and checked the full text. (Line 496)

We also made some changes to the paper. Please see the revised manuscript for details.

Looking forward to your reply.

Best regards,

Chunwang Lai

Institute of Horticultural Biotechnology, Fujian Agriculture and Forestry University, Fuzhou, 350002

E-mail: laichunwang@163.com

Round 2

Reviewer 2 Report

Consideration of the recommendations significantly improved the manuscript.

Please change in the title: line 6: “genes expression level”  to “gene expression level”

I still recommend the authors to ask a native English-speaking person to correct the manuscript.

The figures are still small and hardly readable, however, as you wrote the editors will take care of that.
